# Learning by Doing: The Use of Distance, Corners and Length in Rewarded Geometric Tasks by Zebrafish (*Danio rerio*)

**DOI:** 10.3390/ani11072001

**Published:** 2021-07-05

**Authors:** Greta Baratti, Angelo Rizzo, Maria Elena Miletto Petrazzini, Valeria Anna Sovrano

**Affiliations:** 1CIMeC, Center for Mind/Brain Sciences, University of Trento, 38068 Rovereto, Italy; 2School of Natural Sciences, University of Torino, 10124 Torino, Italy; angelo.rizzo691@edu.unito.it; 3Department of Biomedical Sciences, University of Padova, 35121 Padova, Italy; mariaelena.milettopetrazzini@unipd.it; 4Department of Psychology and Cognitive Science, University of Trento, 38068 Rovereto, Italy

**Keywords:** navigation, spatial learning, environmental geometry, geometric components, zebrafish

## Abstract

**Simple Summary:**

Geometric navigation allows animals to efficiently move towards essential life-spaces by taking advantage of macrostructural information such as distance, angular magnitude, and length, in relation to left-right positional sense. In natural contexts, these cues can be referred to extensive three-dimensional surfaces such as a slope or a riverbed, thus becoming crucial to orient and find useful supplies. In controlled contexts, it is possible to set apart these components by handling the global shape of the experimental space (rectangular or square) as well, with the aim to specially probe the impact of each of them on navigation behavior of animals, including fishes. The present study aimed at investigating whether a well-known vertebrate, the zebrafish, could learn to encode and retain in memory such metric information (in terms of distances, corners, and lengths) in association with left–right directions, to gain rewards. Our results showed that zebrafish learned to use all these geometric attributes when repeatedly exposed to them, over a period of training, thereby giving strength to the ecological relevance of environmental geometry as a source of spatial knowledge. Generally, the engagement of zebrafish may consent to assess computations underlying large-scale-based navigation, also by drawing targeted comparisons, due to its behavioral, cognitive, and even emotional similarities with mammals.

**Abstract:**

Zebrafish spontaneously use distance and directional relationships among three-dimensional extended surfaces to reorient within a rectangular arena. However, they fail to take advantage of either an array of freestanding corners or an array of unequal-length surfaces to search for a no-longer-present goal under a spontaneous cued memory procedure, being unable to use the information supplied by corners and length without some kind of rewarded training. The present study aimed to tease apart the geometric components characterizing a rectangular enclosure under a procedure recruiting the reference memory, thus training zebrafish in fragmented layouts that provided differences in surface distance, corners, and length. Results showed that fish, besides the distance, easily learned to use both corners and length if subjected to a rewarded exit task over time, suggesting that they can represent all the geometrically informative parts of a rectangular arena when consistently exposed to them. Altogether, these findings highlight crucially important issues apropos the employment of different behavioral protocols (spontaneous choice versus training over time) to assess spatial abilities of zebrafish, further paving the way to deepen the role of visual and nonvisual encodings of isolated geometric components in relation to macrostructural boundaries.

## 1. Introduction

In recent years, the growing interest in the engagement of fish species for neuroscientific purposes has allowed comparative psychologists to deeply investigate cognitive skills and related underlying neural mechanisms of this heterogeneous group of organisms [1,2]. Since there is high variability among underwater ecosystems and habitats, fish naturally face spatial orientation challenges, further showing noteworthy navigation capacities such as the use of compasses [3,4], landmarks [5,6], and cognitive maps [7,8]. Besides these skills, fish have been experimentally observed also in relation to the use of environmental geometric layouts to solve place-finding demands after induced disorientation [9,10,11,12,13,14,15,16,17,18,19,20,21,22,23].

The term “environmental geometry” traditionally refers to macrostructural terrain-like characteristics of life-spaces where animals have to pinpoint worthwhile resources such as food, companions, and safe shelters, for survival. As a consequence, geometric navigation embraces the differential encoding of Euclidean concepts (for instance, “point”, “surface”, “boundary”) that can characterize an enclosed space by which animals have to orient. Such a capacity has been demonstrated to be spread across species, both vertebrates [24,25,26,27,28] and invertebrates [29,30,31,32], thus highlighting the adaptive value of cognitive map and geometric representations over the phylogenetic history of animals.

Cheng’s original investigations with *Rattus norvegicus* [33,34] have brought to light the use of spatial geometry in terms of metric (short/long) and directional sense (left/right) information within rectangular arenas, as well as the effect of behavioral protocols on the integrated use of geometric and nongeometric (i.e., local landmarks) cues. A differential encoding of these two types of information has been observed, especially in relation to the memory system involved: while both in working and reference memory tasks, rats made systematic rotational errors (by confusing the right position with the position at 180° on the arena’s diagonal), only in reference memory tasks could rats combine geometric and nongeometric cues to pick out the place where the goal-object was present, thus solving the two-way spatial symmetry. In other words, the encoding of metric-plus-sense properties of surfaces versus features seems to depend on two dissociable cognitive systems, the former being liable to experience (in the presence of learning) while the latter mainly being driven by spontaneous computations (in the absence of learning).

Differences in disoriented navigation under spontaneous or rewarded experimental conditions have been found also in fishes [22,23], in a similar way to mammals [33,34,35]. The behavioral protocols with fish species provide for two well-validated spatial problems: the “social cued memory task” [16,17,18,23] and the “rewarded exit task” [9,10,12,13,15,20,21,22,23]. The social cued memory task consists in assessing spontaneous choices, where fish are required to approach the location of a social object (i.e., a companion) no-longer present, after a phase of passive disorientation. On the other hand, the rewarded exit task consists in assessing reorientation performances over time, where fish are required to learn the location of one or two exit-corridors to gain appealing rewards (i.e., companions, food, a cozy zone). In both cases, fish have to navigate by taking advantage of quantitative and/or qualitative properties of three-dimensional layouts that are marked out by their physical (visible or not) extended boundaries. Currently, however, we do not fully understand the role of isolated quantitative attributes such as distances, angular magnitudes, and lengths, together with the sensitivity to them, when they are not enclosed within a single, interconnected polygon. Most of all, it is still unknown whether and how learning mechanisms (for instance, repetitive trial-and-error training procedures) could aid to bypass the limits of spontaneous reorientation.

The opportunity to test apart these geometric components within fragmented surface arrays arose proper with the aim to specify what kind of metric representations are crucially involved in navigation by geometry. Lee and colleagues [36] found that young children were able to reorient by computing distances, but not lengths, among wall-like surfaces. Again, children could not use neither freestanding objects nor fragmented corners to face and solve disoriented navigation demands [37,38,39,40]. Focusing on nonhuman species, an effect of behavioral protocols has been found in *Gallus gallus*: while trained chicks learned to reorient by means of freestanding objects arrays [41,42,43], untrained chicks did not [44]. Likewise in zebrafish (*Danio rerio*), Lee and colleagues [17] observed that naïve fish spontaneously use distance and directional relationships among three-dimensional extended surfaces (“the goal is left/right of the closer/farther wall) to reorient within a rectangular arena but failed to take advantage of either an array of freestanding corners or an array of unequal-length surfaces to search for a no-longer-present goal, thus being unable to use the information supplied by corners and length without some kind of differential reinforcement. Starting from such an evidence, would zebrafish be able to compute both angular information and length, besides distances, for place-finding after disorientation? Additionally, the use of zebrafish as a model may provide a powerful tool to address navigation issues by applying a combinational approach [45] targeted to link research from multiple neuroscientific fields at both behavioral and neural levels of investigation [46,47]. Nonetheless, the role of zebrafish in multilevel scientific research is currently accepted and established, due to its characteristics and versatility. From pharmacology [48,49,50,51], to genomics [52,53,54], to behavioral studies as well [47], this teleost is becoming potentially useful, even with respect of its emotional and cognitive phenotype.

The purpose of the present work was to investigate whether zebrafish could learn to use the geometric components characterizing a rectangular enclosure, in terms of distance, corners, and length, to gain a reward. Therefore, we adapted the experimental conditions by Lee and colleagues [17] within an apparatus previously used to test both spontaneous and rewarded extra-visual encodings of pure geometry in three-eyed fishes [23]. We performed four experiments, two of them in a rectangular transparent arena, whereas the other two in a square transparent arena. More in details, the four experiments were scheduled as follows: Experiment 1 (“Use of distance as a geometric cue within a rectangular transparent arena”); Experiment 2 (“Use of corners as a geometric cue within a rectangular transparent arena”); Experiment 3 (“Use of length as a geometric cue within a square transparent arena”); Experiment 4 (“Control condition within a square transparent arena”).

All the experiments employed a behavioral protocol widely used in the last 20 years to investigate navigation by geometry in fishes [9,10,12,13,15,20,21,22,23]. Briefly, it consists in training animals over time to locate one or two goal-exits inside a geometrically characterized arena to get a reward, a procedure named “rewarded exit task”. Thus, in order to assess the learning performance of zebrafish, we compared, together with the percentages of choice towards the correct-geometry diagonal versus the incorrect-geometry diagonal. If the opportunity to experience spatial geometric contingencies is a factor to be taken into account for reorientation, then we expected that fish could learn to encode isolated metric components when consistently exposed to them. 

## 2. Materials and Methods

### 2.1. Subjects and Housing

Subjects were 30 wild-type mature male zebrafish (*D. rerio*), ranging from 4 to 5 cm in body-length and coming from breeding stocks in our laboratory. Eight fish were engaged in Experiment 1, Experiment 2, and Experiment 3, while six fish were engaged in Experiment 4. All subjects were naïve, and each of them was observed in only one condition. To excite and catch the attention of the experimental fish, we used two female companions as sexual and social stimuli [55]. Outside of training, fish were maintained under a 10:14-h light/dark cycle and raised within glass home tanks (25 L capacity). Each tank was subdivided in four compartments with the aim to individually breed the experimental subjects. These tanks were further enriched with gravel and plants, thus cleaned with suitable hang-on filters (Niagara 250, Wave) to ensure comfortable habitats. The water temperature was maintained at 26 ± 1 °C by means of 25-watt heaters (NEWA Therm^®^, NEWA, Padua, Italy). Fish were fed with dry food (GVG-Mix Nature, sera^®^ GmbH, Munich, Germany) exclusively within the experimental apparatus from Monday to Friday (the 5-day weekly period of training), to keep the motivation as high as possible.

### 2.2. Experimental Apparatus

The apparatus was the same used by Sovrano and colleagues [23] to test both spontaneous and rewarded extra-visual encodings of pure geometry in three-eyed fishes, where we introduced some adaptations with the aim to replicate the experimental conditions by Lee and colleagues [17], but by employing a different behavioral protocol (see below for details).

In general, the apparatus was placed in a darkened room and consisted of a circular amaranth tank (diameter × height: 175 × 27 cm) laying on a turntable that allowed the experimenter to rotate the whole structure at the beginning of each trial. This tank was surrounded by a circular black curtain fixed on a wood and metal frame, and centrally lit from above (height: 100 cm) through a 24-watt fluorescent white light tube (Lumilux, Osram GmbH, D, Munich, Germany). The water temperature was maintained at 26 ± 1 °C by means of a 50-watt heater (NEWA Therm^®^, NEWA, Padua, Italy), while a filter (NEWA Duetto^®^, NEWA, Padua, Italy) made sure a good quality. Both the heater and filter were not present during the training sessions. 

In the center of such a tank, one of two experimental arenas could be placed: a rectangular transparent arena or a square transparent arena, the former for Experiment 1 and 2, whereas the latter for Experiment 3 and 4. As regards the rectangular transparent arena, it was the same used by Sovrano and colleagues [23], and consisted of an enclosure made by glass (length × width × height: 30 × 20 × 8 cm), thus composed of two long walls (length × height: 26 × 11 cm) and two short walls (length × height: 16 × 11 cm). In order to create a totally nonvisible environment, we did not glue the four walls together, but instead they were inserted in “single-track” supports made by polyvinyl chloride (PVC) and fixed on a Plexiglas base (length × width: 50 × 50 cm) at the bottom of the circular tank, then covered with a thick layer of homogeneous dark gravel (depth: 5 cm). As regards the square transparent arena, it was built ex novo for the purpose of this study and consisted of an enclosure made of glass (length × width: 30 × 30 cm; height: 8 cm), thus composed of four walls of equal size (length × height: 26 × 11 cm). As the rectangular enclosure, the square one stood on a basement (length × width: 50 × 50 cm), while its walls were installed within single-track supports then covered with gravel.

Both arenas were equipped with four “tunnels of choice” (conventionally, “corridors”) placed at the meeting point of each pair of tangent walls. These corridors were positioned towards the outside of the transparent enclosures at precisely 2.5 cm from corners. More in detail, each corridor was composed of three parts: a rectangular transparent glass (length × height: 3 × 11 cm) and two rectangular transparent acetate sheets (length × height: 2.5 × 9 cm). These flexible sheets were perpendicularly glued on the glass in order to create a C-shape design, and they were characterized by a specific pattern of three-series vertical fissures depending on the scope. “Traversable” corridors had to allow fish to exit the arena to gain rewards (i.e., a cozier outer zone provided with food and companions); thus, they had a thick central fissure (length × height: 1 × 7.43 cm) and two thin lateral fissures (length × height: 0.2 × 7.43 cm). Conversely, “nontraversable” corridors had to not allow fish to exit the arena; thus, they had a 3 × 3 matrix of thin fissures (superior and inferior series: length × height: 0.3 × 2.5 cm; central series: length × height: 0.3 × 2 cm). Detail of corridors are depicted in Figure 1.

These slight differences in size were needed to balance the overall perimeter among traversable and nontraversable corridors (47.4 cm) by equalizing hydrodynamic effects due to movements of fish against the physical walls and potentially detectable through the lateral line [23]. For both the arenas, two traversable corridors were placed on the rewarding diagonal (correct-geometry: C_1,2_), whereas two nontraversable corridors were placed on the opposite nonrewarding diagonal (incorrect-geometry: X_1,2_). To be specific, the correct corners were labelled “C_1_” and “C_2_”, while the incorrect ones “X_1_” (close to C_1_) and “X_2_” (close to C_2_). The arenas were entirely inside water, but not submerged (with a gap of 0.5 cm in respect of their height), in order to create visual continuity without light reflections.

With the aim to investigate if fish could learn to use visual geometric cues in terms of distance, length, and corners, for the first three experiments, we added different features made of white polypropylene (Poliplak^®^, Röhm Italia SRL, Milan, Italy): four panels equal in length (length × height: 15 × 10 cm) for Experiment 1; four fragmented corners composed of three panels (central panel: length × height: 3 × 10 cm; lateral panels: length × height: 6 × 10 cm) for Experiment 2; four panels of 2:1 ratio in length (long panels: length × height: 20.4 × 10 cm; short panels: 10.6 × 10 cm) for Experiment 3. Experiment 4 was run in the absence of any additional visual geometric cues, in order to validate the square transparent arena that we built ex novo to replicate the experimental conditions by Lee and colleagues [17], but by applying a different behavioral protocol. The two arenas equipped for all the experiments are depicted in Figure 2.

### 2.3. Experimental Procedure

The training procedure was the same used by Sovrano and colleagues for the nonvisual geometric task with prolonged experience [23], and by Baratti and colleagues to test the encoding of visual geometry in zebrafish [21]. It consisted of a training subdivided in a maximum of 10 daily sessions (one per day), from Monday to Friday, for two successive weeks of work. Each daily session of training consisted in eight trials, until fish reached a learning criterion ≥ 70% of correctness for diagonal C_1,2_ in two consecutive sessions (one per day), the former as a measure of learning achieved and the latter as a validation of performance (accuracy). In case fish were able to reach the criterion within the first two sessions, the training was protracted for two additional sessions (as confirmation of learning). One group of fish was locally rewarded on diagonal 1, while one other group on diagonal 2, aiming to balance potential biases associated to intra/extra-arena cues. 

At the beginning of each trial, fish were gently transferred from their home tank into the experimental apparatus, more precisely within a glass cylinder (diameter × height: 6 × 8) that was placed in the center of the transparent arena. At the upper end of such a cylinder, there was a transparent nylon wire that, by means of a rigid and nonvisible metal pulling mechanism, allowed the experimenter to vertical lift upward the cylinder without creating biases during this tricky operation. After 30 s of acclimation and visual exploration of the environment, the cylinder was slowly lifted up, leaving fish free to swim around and familiarize with the experimental space. Within each trial, fish had a 10-min limit to make choices targeted to the corridors: these attempts were sequentially noted down until fish correctly exited the arena. A correction method was used [56]: fish were allowed to change one or more wrong choices (i.e., towards corridors X_1_ and X_2_), up until they were able to choose one of the two right corridors (i.e., C_1_ or C_2_) or up until the 10-min limit was elapsed. Intervals among trials, where fish could stay within the cozier outer zone to enjoy the rewards, were managed as follows: six minutes (completely rewarding: small amount of food and companions) if fish identified the correct-geometry diagonal C_1,2_ as single attempt, which allow it to get out of the arena, or two minutes (nonrewarding: neither food nor companions) if fish made two or more wrong attempts. Multiple choices for the correct corridors could occur, particularly when fish approached them without exiting of the arena. An attempt was considered as valid if fish entered the corridor with their whole body length; moreover, actual exit attempts were clearly visible in video recordings on the basis of characteristic tail-and-body movements made by fish against the acetate sheets. Whenever a fish did not make a choice within the 10-min limit, it was given a 5-min break inside the outer zone. After two null trials without valid choices, the training session could be interrupted and rescheduled for the day next. Before starting the following trial, the apparatus was rotated 90° right in order to remove any potential uncontrolled cues and reduce the use of compass and inertial information.

### 2.4. Statistical Analysis

With respect to Experiments 1, 2, and 3, we measured the following dependent variables: the mean number of trials to reach a criterion ≥ 70% of correctness in two consecutive learning sessions (learning and validation), together with the percentages of choice towards the correct-geometry diagonal C_1,2_ (thus, by cumulating the percentages of choice towards corners C_1,2_) versus the incorrect-geometry diagonal X_1,2_ (thus, by cumulating the percentages of choice towards corners X_1,2_). With respect to Experiment 4, we inspected the percentages of choice towards C_1,2_ versus X_1,2_ over time, that is, within the 10 sessions scheduled to finalize the training. In the absence of geometry, in terms of metric (short/long) and sense (left/right), the correctness was referred to the traversable corridors placed on the rewarding diagonal of the square arena.

For the first three experiments, Student’s *t*-test for unpaired samples was applied to bear out the lack of cues locally associated to the arenas’ diagonals by comparing the group of fish that was trained on diagonal 1 versus 2 as the correct-geometry C_1,2_. Student’s t-test for paired samples was further applied to compare the percentages of choice towards the correct-geometry diagonal C_1,2_ versus the incorrect-geometry diagonal X_1,2_, with the aim to corroborate the idea that fish had actually learned how to solve the reorientation task. Moreover, Student’s *t*-test for paired samples was applied to compare the two corners laying on the same diagonal (correct-geometry: C_1_ versus C_2_; incorrect-geometry: X_1_ versus X_2_). On the other hand, for the last experiment, a repeated measures ANOVA was applied to compare the percentages of choice towards C_1,2_ versus X_1,2_ over time (1–10 training sessions).

The Shapiro–Wilk test was performed to verify the normality, whereas Levene’s test of equality of error variances and Mauchly’s sphericity test were performed to assess the homoscedasticity. To estimate the effect size of significant data analysis, we reported 95% confidence intervals as an index for Student’s *t*-test. Raw data were analyzed by means of IBM^®^ SPSS Statistic 27 software package, and they are available in a submitted Appendix A.

## 3. Results

### 3.1. Experiment 1: Use of Distance as a Geometric Cue within a Rectangular Transparent Arena

Experiment 1 aimed at investigating if zebrafish could learn to use distance as a geometric cue to reorient in association with directional sense (distance-plus-sense relationships).

The mean number of trials needed to reach the 70% criterion for the correct-geometry diagonal C_1,2_ was 39.625 ± 6.276 (mean ± SEM), which was ≈ 5 consecutive training sessions. Since one group of fish was locally rewarded on diagonal 1 and the other on diagonal 2 to balance any potential uncontrolled biases related to intra/extra-arena cues, the mean number of trials to 70% of correctness for diagonal 1 versus 2 was compared by performing a *t*-test for unpaired samples, which did not reveal significant differences (*t*(6) = 0.927, *p* = 0.390). For this reason, the following analyses were applied by collapsing the two samples.

Results of the learning session, where fish obtained a spatial performance ≥ 70%, and results of the validation session, where fish kept an above-threshold behavior, are shown in Figure 3.

A *t*-test for paired samples applied on both the percentages of first and the total choices in the learning session revealed a significant effect of Geometry (C_1,2_ versus X_1,2_) (first choices: *t*(7) = 14.150, *p* ≤ 0.001, 95% IC [51.362, 71.973]; total choices: *t*(7) = 11.471, *p* ≤ 0.001, 95% IC [41.485, 63.029]). There were no differences between the two corners laying on the same diagonal, by independently comparing the correct-geometry C_1,2_ (first choices: *t*(7) = 0.241, *p* = 0.817; total choices: *t*(7) = −0.254, *p* = 0.807), and the incorrect-geometry X_1,2_ (first choices: *t*(7) = 2.165, *p* = 0.067; total choices: *t*(7) = 2.064, *p* = 0.078).

In a similar way, a *t*-test for paired samples applied on both the percentages of first and total choices in the validation session revealed a significant effect of Geometry (C_1,2_ versus X_1,2_) (first choices: *t*(7) = 6.769, *p* ≤ 0.001, 95% IC [39.039, 80.960]; total choices: *t*(7) = 8.365, *p* ≤ 0.001, 95% IC [43.323, 77.467]). There were no differences between the two corners laying on the same diagonal, by independently comparing the correct-geometry C_1,2_ (first choices: *t*(7) = 0.346, *p* = 0.740; total choices: *t*(7) = 0.056, *p* = 0.957), and the incorrect-geometry X_1,2_ (first choices: *t*(7) = −0.500, *p* = 0.632; total choices: *t*(7) = 0.228, *p* = 0.826).

Results showed that in the presence of nonvisual geometric surfaces (i.e., the rectangular transparent arena) that were visually defined by distance (i.e., the four white panels equal in length), zebrafish easily learned to reorient on the basis of such a visual cue, there perceived in terms of the physical space between them and the four walls, within the 10-session limit allowed to finalize the training. In fact, the percentages of choice towards the correct-geometry diagonal C_1,2_ versus the incorrect-geometry one X_1,2_ statistically departed from chance level (=50%), both in the learning and validation sessions, thus highlighting that fish were able to efficiently navigate to pick out the rewarded corridors by taking advantage of distance-plus-sense spatial relationships.

### 3.2. Experiment 2: Use of Corners as a Geometric Cue within a Rectangular Transparent Arena

Experiment 2 aimed at investigating if zebrafish could learn to use corners as a geometric cue to reorient in association with directional sense (corners-plus-sense relationships).

The mean number of trials needed to reach the 70% criterion for the correct-geometry diagonal C_1,2_ was 60.375 ± 5.568 (mean ± SEM), which was ≈8 consecutive training sessions. Since one group of fish was locally rewarded on diagonal 1 and the other on diagonal 2 to balance any potential uncontrolled biases related to intra/extra-arena cues, the mean number of trials to 70% of correctness for diagonal 1 versus 2 was compared by performing a *t*-test for unpaired samples, which did not reveal significant differences (*t*(6) = −1.693, *p* = 0.141). For this reason, the following analyses were applied by collapsing the two samples.

Results of the learning session, where fish obtained a spatial performance ≥ 70%, and results of the validation session, where fish kept an above-threshold behavior, are shown in Figure 4.

A *t*-test for paired samples applied on both the percentages of first and the total choices in the learning session revealed a significant effect of Geometry (C_1,2_ versus X_1,2_) (first choices: *t*(7) = 14.705, *p* ≤ 0.001, 95% IC [43.533, 60.217]; total choices: *t*(7) = 18.772, *p* ≤ 0.001, 95% IC [45.813, 59.017]). There were no differences between the two corners laying on the same diagonal, by independently comparing the correct-geometry C_1,2_ (first choices: *t*(7) = −0.473, *p* = 0.651; total choices: *t*(7) = 0.130, *p* = 0.900), and the incorrect-geometry X_1,2_ (first choices: *t*(7) = −0.060, *p* = 0.954; total choices: *t*(7) = −0.305, *p* = 0.769).

In a similar way, a *t*-test for paired samples applied on both the percentages of first and total choices in the validation session revealed a significant effect of Geometry (C_1,2_ versus X_1,2_) (first choices: *t*(7) = 8.355, *p* ≤ 0.001, 95% IC [45.813, 80.991]; total choices: *t*(7) = 10.013, *p* ≤ 0.001, 95% IC [50.213, 81.262]). There were no differences between the two corners laying on the same diagonal, by independently comparing the correct-geometry C_1,2_ (first choices: *t*(7) = 1.391, *p* = 0.207; total choices: *t*(7) = 2.130, *p* = 0.071), and the incorrect-geometry X_1,2_ (first choices: *t*(7) = 0.135; *p* = 0.897; total choices: *t*(7) = 0.751, *p* = 0.477).

Results showed that in the presence of nonvisual geometric surfaces (i.e., the rectangular transparent arena) that were visually defined by their corners (i.e., the four white angular panels), zebrafish easily learned to reorient on the basis of these cues within the 10-session limit, allowing the training to be finalized. In fact, the percentages of choice towards the correct-geometry diagonal C_1,2_ versus the incorrect-geometry one X_1,2_ statistically departed from chance level (=50%), both in the learning and validation sessions, thus highlighting that fish were able to efficiently navigate to pick out the rewarded corridors by taking advantage of corners-plus-sense spatial relationships.

### 3.3. Experiment 3: Use of Length as a Geometric Cue within a Square Transparent Arena

Experiment 3 aimed at investigating if zebrafish could learn to use length as a geometric cue to reorient in association with directional sense (length-plus-sense relationships).

The mean number of trials needed to reach the 70% criterion for the correct-geometry diagonal C_1,2_ was 48.750 ± 10.007 (mean ± SEM), which was ≈6 consecutive training sessions. Since one group of fish was locally rewarded on diagonal 1 and the other on diagonal 2 to balance any potential uncontrolled biases related to intra/extra-arena cues, the mean number of trials to 70% of correctness for diagonal 1 versus 2 was compared by performing a *t*-test for unpaired samples, which did not reveal significant differences (*t*(6) = 1.088, *p* = 0.318). For this reason, the following analyses were applied by collapsing the two samples.

Results of the learning session, where fish obtained a spatial performance ≥ 70%, and results of the validation session, where fish kept an above-threshold behavior, are shown in Figure 5.

A *t*-test for paired samples applied on both the percentages of first and the total choices in the learning session revealed a significant effect of Geometry (C_1,2_ versus X_1,2_) (first choices: *t*(7) = 13.899, *p* ≤ 0.001, 95% IC [44.605, 62.895]; total choices: *t*(7) = 15.918, *p* ≤ 0.001, 95% IC [49.229, 66.406]). There were no differences between the two corners laying on the same diagonal, by independently comparing the correct-geometry C_1,2_ (first choices: *t*(7) = −1.193, *p* = 0.272; total choices: *t*(7) = −0.698, *p* = 0.508), and the incorrect-geometry X_1,2_ (first choices: *t*(7) = 0.174, *p* = 0.867; total choices: *t*(7) = −0.390, *p* = 0.708).

In a similar way, a *t*-test for paired samples applied on both the percentages of first and total choices in the validation session revealed a significant effect of Geometry (C_1,2_ versus X_1,2_) (first choices: *t*(7) = 9.029, *p* ≤ 0.001, 95% IC [43.825, 74.925]; total choices: *t*(7) = 9.092, *p* ≤ 0.001, 95% IC [45.833, 78.052]). There were no differences between the two corners laying on the same diagonal, by independently comparing the correct-geometry C_1,2_ (first choices: *t*(7) = 0.341, *p* = 0.743; total choices: *t*(7) = 0.080, *p* = 0.939), and the incorrect-geometry X_1,2_ (first choices: *t*(7) = 0.814; *p* = 0.442; total choices: *t*(7) = −1.014, *p* = 0.344).

Results showed that in the presence of nonvisual nongeometric surfaces (i.e., the square transparent arena) that were geometrically and visually defined by their length (i.e., the four white panels of 2:1 ratio in length), zebrafish easily learned to reorient on the basis of these cues within the 10-session limit allowed to finalize the training. In fact, the percentages of choice towards the correct-geometry diagonal C_1,2_ versus the incorrect-geometry one X_1,2_ statistically departed from chance level (=50%), both in the learning and validation sessions, thus highlighting that fish were able to efficiently navigate to pick out the rewarded corridors by taking advantage of distance-plus-sense spatial relationships.

### 3.4. Control Condition within a Square Transparent Arena

Experiment 4 was a control condition aimed at validating the square arena that we had built ex novo for the purpose of the present study.

Fish did not reach the 70% criterion for the correct-geometry diagonal C_1,2_ within the 10-session limit allowed to finalize the training. Results of training over time (1–10 sessions) are shown in Figure 6.

A repeated measures ANOVA was performed by considering all data collected per fish towards the two diagonals (C_1,2_ versus X_1,2_) throughout the summed 10 sessions of training. The ANOVA with Geometry (C_1,2_ versus X_1,2_) and Session (1–10) as within-subject factors, and Diagonal (1 versus 2) as between-subject factor, then applied on both the percentages of first and total choices, did not reveal any significant differences (first choices: Geometry: F(1, 4) = 1.232, *p* = 0.329; Geometry × Diagonal: F(1, 4) = 1.962, *p* = 0.234; Session: F(9, 36) = 0.949, *p* = 0.496; Session × Diagonal: F(9, 36) = 0.952, *p* = 0.494; Geometry × Session: F(9, 36) = 0.692, *p* = 0.712; Geometry × Session × Diagonal: F(9, 36) = 0.365, *p* = 0.944; Diagonal: F(1, 4) = 1.001, *p* = 374) (total choices: Geometry: F(1, 4) = 5.590, *p* = 0.077; Geometry × Diagonal: F(1, 4) = 2.442, *p* = 0.193; Session: F(9, 36) = 0.953, *p* = 0.493; Session × Diagonal: F(9, 36) = 0.952, *p* = 0.494; Geometry × Session: F(9, 36) = 1.098, *p* = 0.389; Geometry × Session × Diagonal: F(9, 36) = 0.587, *p* = 0.799; Diagonal: F(1, 4) = 1.000, *p* = 0.374).

Results showed that in the presence of nonvisual nongeometric surfaces (i.e., the square transparent arena) that were neither geometrically nor visually defined by some kind of additional cues such as distance (closer/farther), corners (the meeting point of a pair of tangent surfaces), and length (shorter/longer), together with directional sense (left/right), zebrafish did not learn to reorient within the 10-session limit allowed to finalize the training. In fact, the percentages of choice towards the correct-geometry diagonal C_1,2_ versus the incorrect-geometry one X_1,2_ statistically did not depart from chance level (=50%), thus highlighting that fish were totally disoriented in the course of their navigation inside the square transparent arena.

## 4. Discussion

The present study aimed at studying what kind of spatial relationships could be learned by zebrafish during disoriented navigation within a space characterized by three-dimensional extended surfaces. To be specific, we wanted to investigate whether zebrafish could learn to use the geometric components typically characterizing a rectangular-shaped enclosure, in terms of distance (between the center and the walls), corners (as the meeting point of a pair of tangent walls), and length (as the pure metric information short/long), in association with directional sense (left/right). In order to set apart these three components, we required two transparent arenas (i.e., rectangular and square) equipped with distinctive opaque feature (i.e., white polypropylene panels). Starting from evidence by Lee and colleagues [17], where they found that zebrafish were able to spontaneously use only distance-plus-sense relationships to reorient at the expense of corners and length, we wondered if the behavioral protocol they adopted (social cued memory task) might have somehow affected the spatial performance. In this view, two studies by Sovrano and colleagues [22,23] have clearly showed a differential effect of spontaneous and rewarded protocols on geometric navigation of fish.

Therefore, we adapted the experimental conditions by Lee and colleagues [17] within an experimental apparatus previously used to test both spontaneous and rewarded extra-visual encodings of pure geometry in three-eyed fishes [23]. Four experiments were scheduled and then carried out by employing a behavioral protocol widely spread in the last 20 years to investigate navigation by geometry in fishes [9,10,12,13,15,20,21,22,23]. Briefly, it consists in training animals over time to locate one or two goal-exits inside a geometrically characterized arena to get something enticing (food and companions), a procedure named rewarded exit task [22,23]. Fish were required to reach a learning criterion ≥ 70% of correctness in two successive training sessions towards the correct-geometry diagonal of the arena, which they had to identify by encoding the geometric information specially provided by the environment.

Results of Experiment 1 showed that zebrafish easily learned to use the distance component, in terms of the physical space between their starting position in the center of the arena and the four extended surfaces, in order to reorient within a rectangular enclosure that was geometrically defined by visible, white-colored panels equal in length (15 cm). Such evidence strictly replicates findings by Lee and colleagues [17], where they observed that untrained zebrafish spontaneously used distance-plus-sense spatial relationships when subjected to a social cued memory task. In brief, experimental fish were placed inside a glass cylinder in the center of a completely bounded arena that was equipped with four white panels with the same length. At each corner, there was a glass jar (in total, four) where one of them hosted a rewarding sexual companion. Under these conditions, fish were first required to observe and memorize the goal-position, and then to approach it in the absence of the attractor on the basis of the distance component provided by the environment. Zebrafish exhibited a spontaneous tendency for the goal-position, thus making rotational errors targeted to the position at 180° on the diagonal as well. 

Another study by Lee and colleagues with children [36] gave strength to the “inborn” predisposition to use the oneself-surface distance, since human babies (partially supplied with speech, abstract symbols, and cognitive maps) were also able to encode such a component during disoriented navigation within different geometric layouts. In this way, it is plausible that the encoding of distance-plus-sense spatial relationships (i.e., the goal is on the left/right of the closer/farther wall) could be both spontaneous and acquired over experience when animals have to face with place-finding demands. With the purpose to clarify the role of either instinct or learning, neurobiological investigations would be needed, as it has been observed by O’Keefe and Burgess with rats [57], or by comparing the behavior of fish larvae versus adults.

Results of Experiment 2 showed that zebrafish easily learned to use the corners component, in terms of the meeting point of a pair of tangent extended surfaces, to reorient within a rectangular enclosure that was geometrically defined by visible, white-colored panels resembling corners. Such evidence does not replicate findings by Lee and colleagues [17], wherein they observed that untrained zebrafish did not spontaneously use corners-plus-sense spatial relationships if subjected to a social cued memory task (working memory).

In the case of the rewarded exit task (reference memory), the behavioral protocol exerted a relevant part on spatial performances of fish: by consistently exposing them to a fragmented array of corners, *de facto*, zebrafish were able to infer the macrostructural morphology of the rectangular arena to efficiently reorient. Despite the absence of an “occludent”, such a situation may be compared to the “amodal completion” phenomenon, since there is a kind of reconstruction of surfaces without direct stimulation of vision (i.e., without object-to-retina physiological correspondence). The amodal completion effect has been observed in fish, particularly with respect to bidimensional shape discrimination, both in controlled settings [58] and in natural contexts [59]. In this vein, such a perception could play a role as for goal-oriented navigation by zebrafish, which typically inhabit muddy environments where the exposure to partially covered objects is recurrent.

Results of Experiment 3 showed that zebrafish easily learned to use the length component, in terms of the pure metric information refereed to the extended surfaces (short/long), to reorient within a square enclosure that was geometrically defined by visible, white colored panels of 2:1 ratio in length. Such evidence does not replicate findings by Lee and colleagues with humans [37,38,39,40] and zebrafish [17], wherein they observed that untrained people and fish did not spontaneously use length-plus-sense spatial relationships if subjected to working memory conditions. On the contrary, we demonstrated that zebrafish are able to take advantage of length-plus-sense spatial relationships (i.e., the goal is on the left/right of the shorter/longer wall) if subjected to reference memory conditions. Analogously to Experiment 2, it seems that the chance to be exposed over time to specific geometric contingencies may have aided fish to retain the only attribute available around them, this being the different ratio in length provided by the four opaque panels. Hence, these findings highlight how motivational factors driven by some kind of recompense can affect the ability to represent in mind such a spatial relationship.

Although both untrained two-year-old babies and zebrafish ignore metric information as a cue to reorient within three-dimensional layouts, recent studies have shown that they were susceptible to the length of pictures and small objects [60]. In more detail, by applying a paradigm called “deviant detection paradigm” where five out of six types of “L” shared a geometric feature, children aged 4–10 were able to recognize the “geometric deviant” that diverged with respect to the control stimuli on the basis of their axis’ length. Therefore, a predisposition to encode and make use of this characteristic may exist in certain circumstances [61]. Similarly, it has been observed that guppies (*Poecilia reticulata*) showed a tendency to overestimate the length of vertical stimuli such as, for example, the bisector of a reverse “T”, thereby providing additional support to the capacity to represent metric attributes in humans as well as in aquatic species [62].

Experiment 4 was a control aimed at evaluating whether the square transparent arena we built ex novo to replicate, under training, the experimental condition under spontaneous navigation by Lee and colleagues [17] did not have spurious nongeometric cues thereby affecting spatial performances achieved by fish in Experiment 3. In line with our hypothesis, zebrafish were totally disoriented inside the square transparent arena devoid of any opaque panels, thus further highlighting the actual role of length on reorientation. Nevertheless, the validation of the square transparent arena is relevant also in light of the following experiments targeted to investigate if zebrafish can learn to use salient pattern-specific landmarks at corners in the absence of metric. The idea behind it is to evaluate more in depth if discrimination learning abilities could be eased or hampered by spatial geometric contingencies.

Results of Experiment 4 are at odds with previous findings by Sovrano and colleagues [23], wherein they found that three species of fish (*D. rerio*, *X. eiseni*, and *C. auratus*) learned to reorient within a rectangular transparent arena, thereby in the absence of visibility, by means of extra-visual sensory modalities (for instance, the lateral line or the sense of touch). In fact, fish were able to encode the environment’s shape that was characterized by two short and two long walls only in cases of consistent trial-and-error experience, but not during spontaneous navigation, further supporting the observations by Lee and colleagues with the social cued memory task [17,18]. Because of all this evidence, the key role of behavioral protocols (spontaneous versus rewarded) and, as a consequence, of the memory system mainly recruited (working versus reference), become strongly evident in the solution of spatial reorientation problems.

Altogether, the results of our study allow us to better understand what kind of geometric relationships zebrafish are susceptible to, thereby providing a threefold scientific contribution. First, we confirmed the tendency of this species to use distance-plus-sense relationships, and, in addition, we demonstrated that fish could learn to use corners-plus-sense (through a kind of amodal reconstruction of the nonvisible surfaces) and length-plus-sense as well, in order to pinpoint the two geometrically equivalent positions. Distance, corners, and length are easily available in rectangular visible three-dimensional contexts; in fact, it has been shown that fishes are highly responsive to them [9,10,11,21]. Furthermore, the spatial performance of fish in the four experiments is ascribable to a context where one only geometric component was present: for such a reason, a higher number of training trials to learn may be due to interindividual differences [63]. Second, the present work focuses on the role of protocols (operant conditioning versus spontaneous choices), somehow highlighting that both can determine what sort of abilities animals display when solving high levels cognitive demands. In more depth, we showed the remarkable impact of rewards on goal-oriented behavior of zebrafish, a species that it has been observed to be sensitive to food [64,65] and companions [48,65] as motivational factors. As regards geometric navigation, Sovrano and colleagues [22] observed in *X. eiseni* improvements in landmark-based reorientation under rewarded training over time. In general, the two protocols showed different results because they clearly implement different behavioral techniques to test geometric reorientation capacities of fish. While the social cued memory task focuses on unconstrained approaches made by animals in the absence of a reward that follows a correct choice for the time being, the rewarded exit task takes proper advantage of forced motivation levels (on the basis of operant conditioning rules), which probably undergo an increment over time (e.g., the more days fish are “hungry and alone”, the more they will search for food and companions). Indeed, for future similar studies with zebrafish, it would be interesting to deepen the impact of training on disoriented navigation by replicating another study by Lee and colleagues in working memory [18]. The broad idea could be to adapt in reference memory all the experimental conditions where fish failed to reorient by means of nonvisual quantitative and visual qualitative cues (i.e., the use of 2D form, 3D proximal landmark, proximal light source, distal landmark) in order to better clarify the effect of persistent incentives on such a spatial skill. Third, our results underline the role of memory as independent systems (short versus long term) on the encoding of isolated metric attributes. Specifically, how the opportunity to consistently experience macrostructural cues is essential to consolidate and fix distances, corners, and length characterizing geometric frameworks. These findings seem in line with previous studies with zebrafish: while short-term memory had flexibility limits in the solution of certain cognitive tasks [66], long-term memory allowed fish to recall a spatial orientation activity up to 10 days after a test [67].

Even though the ability to reorient by geometry is spread among vertebrates [27,68] and invertebrates [29,30,31,32], the nature of the spatial relationships beneath has been superficially investigated, most of all in humans [36,37,38,39,40]. In fact, such an issue differs from all the studies concerning the integrated use of conspicuous or local landmarks in association with macrostructural geometry to facilitate the solution of reorientation tasks, where spatial performances of fishes are comparable with those achieved by other land tetrapods [27]. In general, boundary-and landmark-based navigation seems to be supported by two independent cognitive mechanisms: a “spatial system”, for the processing of invariable geometric information, and a “landmark system”, for the processing of variable nongeometric information [69]. Since homologies between the hippocampus of mammals and the lateral pallium of teleosts have been established [70,71], together with a common organization of the basal ganglia [72], it is plausible that three-dimensional frameworks and features may be differently implemented in the brain of vertebrates (hippocampus versus striatum), including fishes. Furthermore, evidence in chicks [73,74] and fish [75] have highlighted the preferential engagement of the right hemisphere in the encoding of extended layouts during geometry-driven navigation by providing additional contribution to such a functional dissociation.

Besides geometric skills, it has been proven that aquatic organisms possess remarkable learning and memory capacities [76,77], which has led to a comprehensive map of life-spaces being built. Fish species are further supplied with alternative nonvisual sensory modalities, as the “lateral line” [78], whose functionalities may be involved in reorientation spatial phenomena in condition of scant visibility (e.g., visual transparency or blindness conditions) [20,23].

## 5. Conclusions

To conclude, our study falls within an intriguing and well-explored topic of spatial cognition, that is, the capacity to navigate by means of the geometric properties provided by an environment with a definite shape in terms of extended surfaces. In more detail, we aimed to better disentangle the metric components characterizing such a context by independently dissecting each of these attributes (distance, corners, and length) in relation to left-right positional sense. Not only that, but we were also interested in understanding whether the methodology we applied could have played a crucial part on learning behavior of disoriented zebrafish. Choosing the most suitable procedure, in fact, is an essential step to study animals’ behavioral patterns, depending on purposes. Whereas spontaneous activity allows us to corroborate the presence of a given skill as a natural trait, the possibility to train animals to a desired behavior may aid in going beyond the constraints imposed by the phylogenetic history of species, with the aim of understanding the extent to which fish can learn to solve a cognitive task and test their learning skills. Lastly, the use of zebrafish as a model may provide a powerful tool to address navigation issues, with the purpose of drawing a phylogenetic line across remote species as regards the computations underlying geometric reorientation. Moreover, the increasingly widespread engagement of such a species in genomics allows for implementation of combinational approaches [45] targeted to investigate spatial cognition skills at every degree of analysis.

## Figures and Tables

**Figure 1 animals-11-02001-f001:**
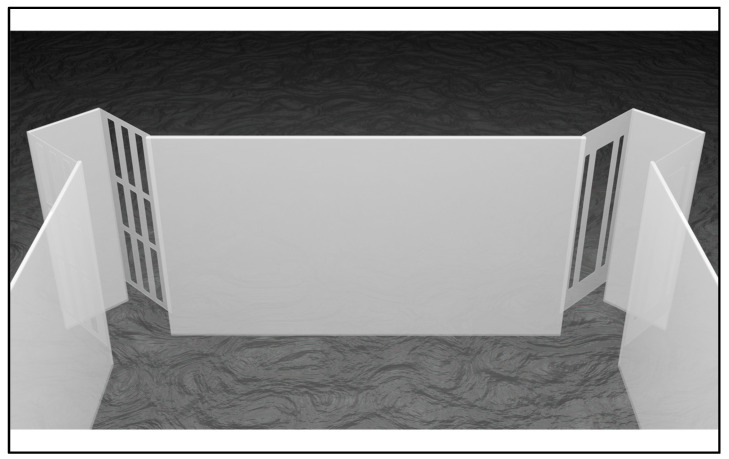
Detail of corridors. At level of the arena’s corners, four corridors (two per side) made of glass and acetate were placed: two traversable corridors having a large central hole were arranged on the correct-geometry diagonal, whereas two nontraversable corridors having smaller holes were arranged on the incorrect-geometry one. While the former allowed fish to exit the arena, thus achieving food and companions, the latter did not.

**Figure 2 animals-11-02001-f002:**
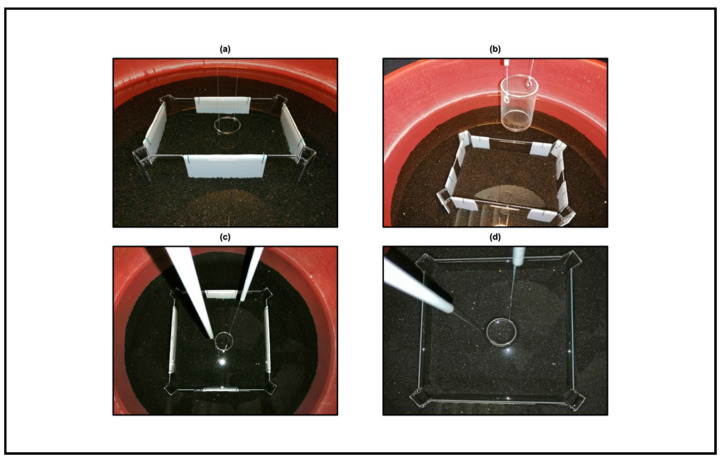
Photographs of the experimental arenas. (**a**) The rectangular transparent arena equipped with four white equal-length panels to assess the distance component. (**b**) The same rectangular arena but provided with four white angular panels to assess the corners component. (**c**) The square transparent arena equipped with four white unequal-length panels of 2:1 ratio to assess the length component. (**d**) The same square arena devoid of any supplementary visual cues to bear out the absence of potential biases associated to intra/extra-arena cues. All the variations were supplied with a transparent cylinder, placed in the center, where experimental fish were hosted before starting each training trials. Such a cylinder was lifted by the experimenter through a pulley system from the outside in order to not be seen by animals.

**Figure 3 animals-11-02001-f003:**
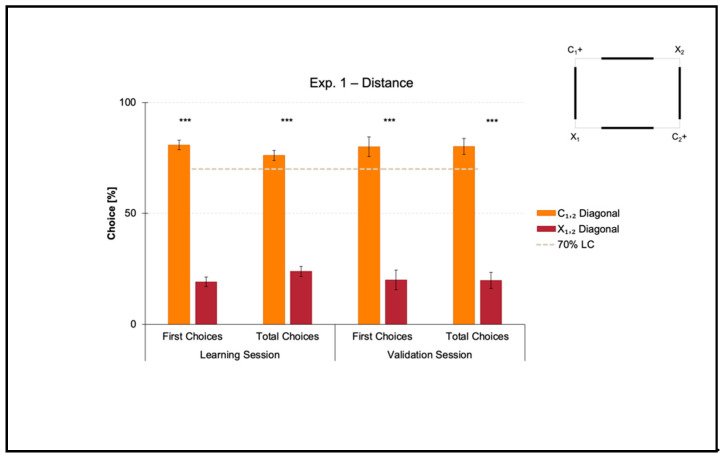
Results of Experiment 1. The bar chart shows the percentages of first and total choices (mean, SEM) obtained by fish towards the two diagonals (C_1,2_ versus X_1,2_) in both learning and validation sessions. The gray dashed line represents the 70% learning criterion (LC), while the asterisks (***) indicate a *p*-value ≤ 0.001.

**Figure 4 animals-11-02001-f004:**
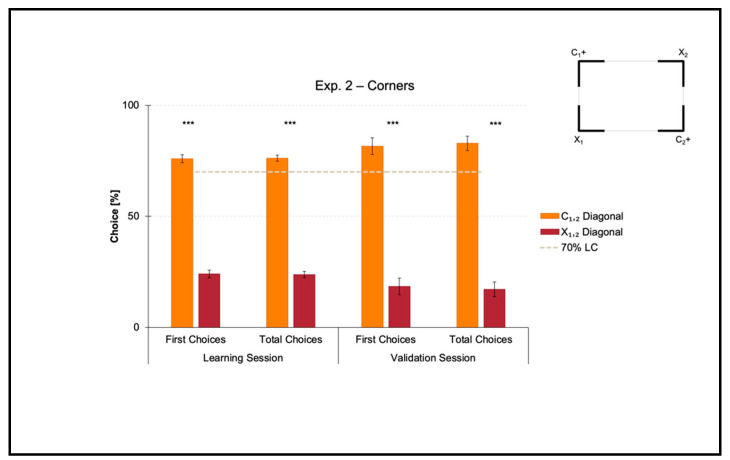
Results of Experiment 2. The bar chart shows the percentages of first and total choices (mean, SEM) obtained by fish towards the two diagonals (C_1,2_ versus X_1,2_) in both learning and validation sessions. The gray dashed line represents the 70% learning criterion (LC), while the asterisks (***) indicate a *p*-value ≤ 0.001.

**Figure 5 animals-11-02001-f005:**
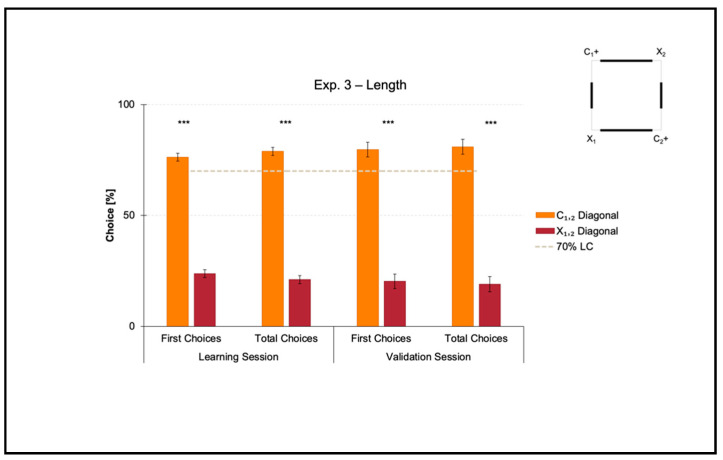
Results of Experiment 3. The bar chart shows the percentages of first and total choices (mean, SEM) obtained by fish towards the two diagonals (C_1,2_ versus X_1,2_) in both learning and validation sessions. The gray dashed line represents the 70% learning criterion (LC), while the asterisks (***) indicate a *p*-value ≤ 0.001.

**Figure 6 animals-11-02001-f006:**
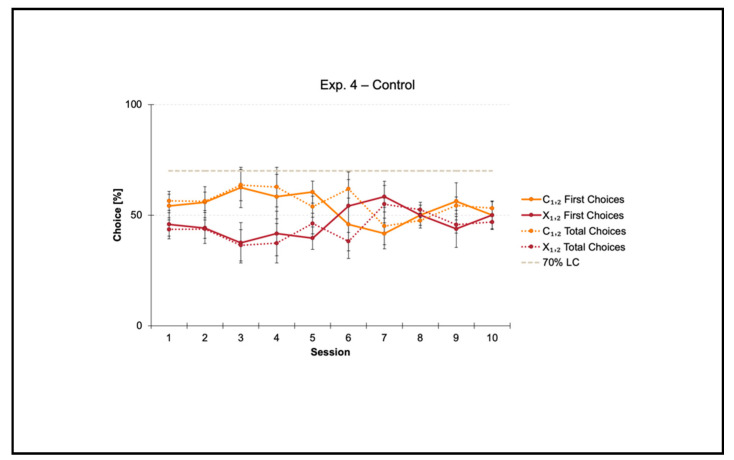
Results of Experiment 4. The line chart shows the percentages of choice (mean, SEM) obtained by fish towards the two diagonals (C_1,2_ versus X_1,2_) over time, that is, for each of the 10 training sessions. Solid lines represent first choices, and dotted lines represent total choices, while the gray dashed line represents the 70% learning criterion (LC).

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
