# Peer review of "Learning by Doing: The Use of Distance, Corners and Length in Rewarded Geometric Tasks by Zebrafish (Danio rerio)"

_animals, 2021, doi:10.3390/ani11072001_

Round 1
Reviewer 1 Report
The paper of Greta Baratti and colleagues focus on a very interesting topic, already widely treated by the same authors during their career as a line of research, as evidenced by the numerous references, however well centered. The study is well conducted, in a very accurate way, to confirm the experience in the topic. My opinion on this manuscript is, for these reason, quite good, it deserves the publication after some minor revisions.
Indeed, I found several points to address before publication, which I summarize as follows:
In first, the manuscript needs an English language editing made by an expert, especially for Simple Summary/Abstract and Introduction sections that at the present form results a little confused and not totally clear in several points. Almost the entire manuscript was written with an extreme care, that if on the one hand it is a positive thing, in the descriptive and introductory parts it risks to confuse the reader and overly burden the reading, decentralizing attention from the salient contents of the work (which are abundant) and not making very clear the concepts that the Authors want to express. My advice is to to be helped by a language expert and to thin out the text in the parts where it is not necessary to go into too much detail, or to insert personal comments in Materials and Methods and Results sections, concentrating them in the Discussions section.
For better understanding what I mean, see for example the lines between:
- 228-241
- 352-354
- 397-399
- 443-445
- 478-484
Personally, I do not very much approve of including in the text, upstream of a reference, expressions like "see for instance" or similar, as it is already clear and not for example, since they are references, that the authors refer to it.
Although it is a widely used and well-established study model, it is worth adding a few sentences to support the role of zebrafish in scientific research, since it is widely used in studies of various nature for its characteristics, for example see and eventually cite this references:
Salvaggio, A., Marino, F., Albano, M., Pecoraro, R., Camiolo, G., Tibullo, D., Bramanti, V., Lombardo, B.M., Saccone, S., Mazzei, V., Brundo, M.V. Toxic effects of zinc chloride on the bone development in Danio rerio (Hamilton, 1822) (2016) Frontiers in Physiology, 7 (APR), art. no. 153. DOI: 10.3389/fphys.2016.00153 Lauriano, E.R., Guerrera, M.C., Laurà, R., Capillo, G., Pergolizzi, S., Aragona, M., Abbate, F., Germanà, A. Effect of light on the calretinin and calbindin expression in skin club cells of adult zebrafish. (2020) Histochemistry and Cell Biology, 154 (5), pp. 495-505. DOI: 10.1007/s00418-020-01883-9Cascio, P.L., Calabrò, C., Bertuccio, C., Iaria, C., Marino, F., Denaro, M.G. Immunohistochemical characterization of PepT1 and Ghrelin in Gastrointestinal tract of zebrafish: Effects of Spirulina vegetarian diet on the neuroendocrine system cells after alimentary stress (2018) Frontiers in Physiology, 9 (MAY), art. no. 614, DOI: 10.3389/fphys.2018.00614
Why are group 4 animals less in experimental (6 and not 8) design than the other three experiments? This fact should be better argued by the Authors.
The manuscript needs a Conclusions section separate from Discussion. At the present form the conclusions are inserted at the end of the Discussion section between the lines 629-634, but, in my opinion, it are too concise and should be expanded by relating them more to the strengths and innovations obtained in the study, giving the right attention that deserves a study of this value, for scientific community.
Best regards
The Reviewer
Author Response
The paper of Greta Baratti and colleagues focus on a very interesting topic, already widely treated by the same authors during their career as a line of research, as evidenced by the numerous references, however well centered. The study is well conducted, in a very accurate way, to confirm the experience in the topic. My opinion on this manuscript is, for these reason, quite good, it deserves the publication after some minor revisions.
Indeed, I found several points to address before publication, which I summarize as follows:
In first, the manuscript needs an English language editing made by an expert, especially for Simple Summary/Abstract and Introduction sections that at the present form results a little confused and not totally clear in several points. Almost the entire manuscript was written with an extreme care, that if on the one hand it is a positive thing, in the descriptive and introductory parts it risks to confuse the reader and overly burden the reading, decentralizing attention from the salient contents of the work (which are abundant) and not making very clear the concepts that the Authors want to express. My advice is to be helped by a language expert and to thin out the text in the parts where it is not necessary to go into too much detail, or to insert personal comments in Materials and Methods and Results sections, concentrating them in the Discussions section. For better understanding what I mean, see for example the lines between:
228-241 352-354 397-399 443-445 478-484
We accepted the reviewer's suggestions. We removed all personal comments from Methods and Results sections, to report them in the Discussion section.
Personally, I do not very much approve of including in the text, upstream of a reference, expressions like "see for instance" or similar, as it is already clear and not for example, since they are references, that the authors refer to it.
Following the suggestion of the reviewer, we deleted these expressions where present in relation to references.
Although it is a widely used and well-established study model, it is worth adding a few sentences to support the role of zebrafish in scientific research, since it is widely used in studies of various nature for its characteristics, for example see and eventually cite this references:
- Salvaggio, A., Marino, F., Albano, M., Pecoraro, R., Camiolo, G., Tibullo, D., Bramanti, V., Lombardo, B.M., Saccone, S., Mazzei, V., Brundo, M.V. Toxic effects of zinc chloride on the bone development in Danio rerio (Hamilton, 1822) (2016) Frontiers in Physiology, 7 (APR), art. no. 153. DOI: 10.3389/fphys.2016.00153
- Lauriano E.R., Guerrera, M.C., Laurà, R., Capillo, G., Pergolizzi, S., Aragona, M., Abbate, F., Germanà, A. Effect of light on the calretinin and calbindin expression in skin club cells of adult zebrafish. (2020) Histochemistry and Cell Biology, 154 (5), pp. 495- 505. DOI: 10.1007/s00418-020-01883-9
- Cascio, P.L., Calabrò, C., Bertuccio, C., Iaria, C., Marino, F., Denaro, M.G. Immunohistochemical characterization of PepT1 and Ghrelin in Gastrointestinal tract of zebrafish: Effects of Spirulina vegetarian diet on the neuroendocrine system cells after alimentary stress (2018) Frontiers in Physiology 9 (MAY) art no 614 DOI:
We did it, by also adding the references mentioned by the reviewer.
Why are group 4 animals less in experimental (6 and not 8) design than the other three experiments? This fact should be better argued by the Authors.
The first three conditions were experimental: fish had to prove they were capable to learn a task. In the control condition (Exp. 4), however, there was not something that could be learned, in fact, there are no results in that direction. In general, we have followed the ethical principle of Reduction, by using the smallest as possible number of animals to obtain statistically significant data analysis. This occurred only in the three experimental conditions where animals were expected to learn a task, as it is happened, but not in Exp. 4, the control condition, where no learning was expected, as it happened. In fact, six out of six animals did not achieve any significant results. No more animals were to be added. Furthermore, the number of animals that we can use in the experiments is strictly established by the Italian Ministry of Health, which approved our research project. We cannot overcome it, precisely in relation to the principle of Reduction. It seemed to us that this aspect was very technical and intuitive and that it did not require additional specifications in the text, to avoid weighing it down.
The manuscript needs a Conclusions section separate from Discussion. At the present form the conclusions are inserted at the end of the Discussion section between the lines 629-634, but, in my opinion, it are too concise and should be expanded by relating them more to the strengths and innovations obtained in the study, giving the right attention that deserves a study of this value, for scientific community.
We followed all the Reviewer’s suggestions, so the manuscript is now enriched by a Conclusions paragraph.
Reviewer 2 Report
This is a very interesting paper, and I congratulate the authors for their effort.
Unfortunately, I feel that the topic is too specific and too theoretical for a journal like "Animals". I would suggest the authors to try with a more behavioural-inclined kind of journal, with an empasis on cogniton and/or behavioural processes.

Author Response
The manuscript has been submitted to the special issue titled “Behavioural Methods to Study Cognitive Capacities of Animals” that is focused on behaviour and cognition in animals. This is way we believe the topic of our manuscript is fully pertinent to the special issue.
Reviewer 3 Report
Dear authors,
The idea to propose new methods to test special location using geometric features is remarkably interesting, however methods, results and discussion need to be improved to convince readers and better fit the scientific demand.
At first, the methods section is difficult to follow due to the amount of information, thus as much visual you can add, the better. For instance, several questions came to me when reading it:
- As it is a rewarding exit task, where was the exit of the tank used? To my view, both squared and rectangle tanks have no exit at the corners and then how fish could access social/sexual reward? Did you use glass jars as used by Lee et al. (2013)? Or did you net the fish out of the tank for the 6min reward?
- The number of trials, sections were also confusing to understand. Did you tested fish in 8 subsequent trials to form one section, and then it was repeated 10 times (10 sections), once every day for 10 days? Does it mean that fish were trained 800 times? (lines 245 to 249)
-Then it says on line 246: “…training phase subdivided in a maximum of ten daily sessions…” Does it mean that not all fish were trained for 10 sections? Actually, you have very small number of animals used in each treatment, so if each one went through a different training, it should be clear stated in the text.
- It may be useful to include a timeline of training and testing
-Animals used were all males. Why? Also, state which strain was used.
- Line 269: what is the reasoning for “punishing” the fish if it only made a correct choice after two or more wrong choices? Why not giving the reward every time fish made a correct choice?
For statistics and results presentation, first, using a Student’s t-test for paired samples to compare correct versus incorrect choice is not valid because the correct choice implies ti was not a incorrect and vice-versa. Thus, you should use a binomial test or a Qui-squared test that evaluates the chance of choosing the correct corner versus the random choice of choosing any other corner. As the correct corner was only one (c1 or c2), it would be 25% chance only. Also, I would like to see your results presented over time (over sections), as you have done for the experiment 4. Additionally, present the values obtained for each fish as a dot instead of presenting mean values, thus one can see animals distribution. You may also want to compare the types of spatial learning used by comparing fish performance in experiment 1, 2 and 3 and including overtime performance, then I suggest you apply a GLMM test (followed by slmeans).
The discussion section needs to be improved. First, I can only see a comparison between fish and humans, what is a huge jump. Please, provide data from other taxa and show how the ability to guide from geometric features in the ambient can be evolved: it is likely that fish present the core system for organizing such informations in a way they use for navigation while derived animals may have other layers of information processing.
I would also like to see more neural basis for the ability, for instance which areas of the brain may be related to special navigation using geometric features of different types.
Then, discuss what it all means and what is the importance to a fish? How it can be used further?
Author Response
The idea to propose new methods to test special location using geometric features is remarkably interesting, however methods, results and discussion need to be improved to convince readers and better fit the scientific demand.
At first, the methods section is difficult to follow due to the amount of information, thus as much visual you can add, the better. For instance, several questions came to me when reading it:
As it is a rewarding exit task, where was the exit of the tank used? To my view, both squared and rectangle tanks have no exit at the corners and then how fish could access social/sexual reward? Did you use glass jars as used by Lee et al. (2013)? Or did you net the fish out of the tank for the 6min reward?
Thanks for this note. We realized that details of the “tunnels of choice” are not visible from the currently present figures. We have added a new figure to better show the “open” and “closed” doors that were placed at the end of corridors.
The number of trials, sections were also confusing to understand. Did you tested fish in 8 subsequent trials to form one section, and then it was repeated 10 times (10 sections), once every day for 10 days? Does it mean that fish were trained 800 times? (lines 245 to 249)
Thank you for this doubt: we realized that the text was missing the specification that “each daily session” (one per day) consisted of 8 training trials. We changed it consistently.
Then it says on line 246: “...training phase subdivided in a maximum of ten daily sessions...” Does it mean that not all fish were trained for 10 sections? Actually, you have very small number of animals used in each treatment, so if each one went through a different training, it should be clear stated in the text.
10 sessions were the maximum period of training each animal was allowed to learn the task. Fish were trained until the learning criterion was reached (≥70% of correct choices), which had to be maintained for two consecutive days. This performance typically occurred within 10 daily sessions. The only constraint, which we have specified in the text, is that if a fish has already learned in the second training session, the training was protracted for two additional sessions, in order to be sure there was an effective learning. We specified it better in the Procedure subsection.
It may be useful to include a timeline of training and testing
After reading this request, we realized that we have incorrectly indicated the term "training-phase" in the text, which we have corrected. The experiment did not include a test-phase after the training, but only a training until the fixed criterion was reached.
Animals used were all males. Why? Also, state which strain was used.
We used only males because females tend to be less responsive, in our experience. On the other hand, it is easier to incentivize males, mainly due to the sexual attractiveness provided by females, as well as food. The strain used was “wild type”. We added the specification in the text.
Line 269: what is the reasoning for “punishing” the fish if it only made a correct choice after two or more wrong choices? Why not giving the reward every time fish made a correct choice?
In general, every time fish made a correct choice, it was able to get out of the training arena. Within a single trial, the fish can exit through one of the correct corridors or make several attempts towards the wrong corridors, before finding the correct exit-door. To make the operant conditioning highly effective, so that fish learn the task as soon as possible, it is essential to specially reinforce those trials where animals make only the correct choice. Hence, the absence of full rewards (not exactly a “punishment”) was used to discourage fish from making multiple wrong choices but rather learn to choose the correct doors right away.
For statistics and results presentation, first, using a Student’s t-test for paired samples to compare correct versus incorrect choice is not valid because the correct choice implies ti was not a incorrect and vice-versa. Thus, you should use a binomial test or a Qui- squared test that evaluates the chance of choosing the correct corner versus the random choice of choosing any other corner. As the correct corner was only one (c1 or c2), it would be 25% chance only. Also, I would like to see your results presented over time (over sections), as you have done for the experiment 4. Additionally, present the values obtained for each fish as a dot instead of presenting mean values, thus one can see animals distribution. You may also want to compare the types of spatial learning used by comparing fish performance in experiment 1, 2 and 3 and including overtime performance, then I suggest you apply a GLMM test (followed by slmeans).
We are comparing the percentages of choices towards the four corners of the arena. The number of overall choices can range from 0 to infinite and cannot be represented on a nominal scale. Moreover, the use of percentages, with an absolute zero, allows us to opt for scales with a greater complexity. These are the analyzes that are consistent with those also present in all the literature cited in the manuscript about geometric reorientation in fish, and that let us to do the most appropriate comparisons with respect to previous evidence. Similarly, the way of representing the results was chosen for comparative purposes; thus, we decided to report the data of the learning sessions only. To us, this choice sounds appropriate, since each fish had its own learning time (thus, an individual learning curve). According to previous consolidate literature, we have reported on charts the mean percentages of the learning times for each experimental condition (Experiments 1, 2, 3). Probably, the line chart of the control condition (Experiment 4) is superfluous, because no animal has learned, and we could have just declared it in the text, without adding charts. This could be the appropriate choice to make. We think this is the easiest reading-mode for the results. Adding charts with individual performances would make it more “tortuous” and difficult to follow, especially if the aim is to consider the overall findings. Finally, comparing the experiments by means of a dedicated analysis does not seem appropriate to us, since the experiments are completely different from each other. Each experiment has its own intrinsic logic and finds its natural comparison with what has been performed by using a paradigm of spontaneous choices (Lee et al., 2013, J Exp Biol). For this reason, we think that the comparison should not be made among the different experiments presented here as they are not different conditions of the same Experiment.
The discussion section needs to be improved. First, I can only see a comparison between fish and humans, what is a huge jump. Please, provide data from other taxa and show how the ability to guide from geometric features in the ambient can be evolved: it is likely that fish present the core system for organizing such informations in a way they use for navigation while derived animals may have other layers of information processing.
We added a new paragraph in the discussion.
I would also like to see more neural basis for the ability, for instance which areas of the brain may be related to special navigation using geometric features of different types.
We did it as suggested.
Then, discuss what it all means and what is the importance to a fish? How it can be used further?
We specified it in the discussion.
Round 2
Reviewer 2 Report
I am positively impressed by the response of the authors' to my comments. The paper, in my opinion, is now suitable for publication in "Animals".